# Gut Microbiota Profiling as a Promising Tool to Detect Equine Inflammatory Bowel Disease (IBD)

**DOI:** 10.3390/ani14162396

**Published:** 2024-08-18

**Authors:** Tiina Sävilammi, Rinna-Riikka Alakangas, Tuomas Häyrynen, Silva Uusi-Heikkilä

**Affiliations:** 1Department of Biological and Environmental Science, University of Jyväskylä, P.O. Box 35, 40014 Jyväskylä, Finland; tiina.m.savilammi@jyu.fi (T.S.); rinna-riikka.r.a.alakangas@student.jyu.fi (R.-R.A.); 2Department of Biology, University of Turku, 20014 Turku, Finland; 3Laukaa Horse Hospital, Ravitie 4, 41330 Vihtavuori, Finland; tuomas.hayrynen@gmail.com

**Keywords:** fecal sample, 16S sequencing, amplicon sequencing, diagnostic marker, horse

## Abstract

**Simple Summary:**

Inflammatory bowel disease (IBD) has become more common among humans and awareness of its prevalence in animals, including horses, has also increased during the past few decades. Equine IBD is problematic as it decreases the performance of horses and causes diarrhea, weight loss, and pain-related behaviors. Apart from high levels of uncertainty in diagnosing IBD, it is stressful for the horse and expensive for the owner as it typically requires transportation to the horse hospital, fasting, endoscopy, and biopsy assessment. Therefore, there is a need for a non-invasive and affordable method to screen for IBD. Fecal microbiota composition has been shown to change in humans and other animals with IBD; therefore, we used fecal samples to study whether equine IBD is also reflected by changes in the composition of gut microbiota. We show that gut microbiota composition was different and that the abundances of certain bacterial groups were significantly increased or decreased in horses with IBD compared to healthy horses. We further used a machine learning model to predict IBD based on the microbiota composition of the fecal samples. The model produced correct predictions with 100% accuracy in our dataset. We conclude that, in the future, studying the microbiota composition of fecal samples could become an accurate, cheap, and non-invasive method in screening for equine IBD.

**Abstract:**

Gastrointestinal disorders are common and debilitating in horses, but their diagnosis is often difficult and invasive. Fecal samples offer a non-invasive alternative to assessing the gastrointestinal health of horses by providing information about the gut microbiota and inflammation. In this study, we used 16S sequencing to compare the fecal bacterial diversity and composition of 27 healthy horses and 49 horses diagnosed with inflammatory bowel disease (IBD). We also measured fecal calprotectin concentration, a marker of intestinal inflammation, in healthy horses and horses with IBD. We found that microbiota composition differed between healthy horses and horses with IBD, although less than five percent of the variation in microbiota composition was explained by individual health status and age. Several differentially abundant bacterial taxa associated with IBD, age, or body condition were depleted from the most dominant Firmicutes phylum and enriched with the Bacteroidota phylum. An artificial neural network model predicted the probability of IBD among the test samples with 100% accuracy. Our study is the first to demonstrate the association between gut microbiota composition and chronic forms of IBD in horses and highlights the potential of using fecal samples as a non-invasive source of biomarkers for equine IBD.

## 1. Introduction

The urban environment alters gut microbiota in both humans and animals [1,2]. Consequentially, the prevalence of many diseases, such as inflammatory bowel disease (IBD), has increased [1,3]. IBD is a shared term for a group of conditions that are characterized by inflammation in the gastrointestinal tract (GIT) and has been diagnosed and studied not only in humans but also in domestic animals, including horses (e.g., [4]). Although IBD research has been utilizing experimental models and advanced technologies, the underlying causes of IBD are still unknown [5,6]. To further our understanding of what causes IBD, research has increasingly been focusing on the association between IBD and gut microbiota composition [4,7,8,9,10].

The gut microbiota consists of the micro-organisms in an individual’s GIT and includes bacteria, fungi, viruses, and archaea. The gut microbiota community is complex and dynamic, and its functions extend beyond digestion, affecting immunity and metabolism and strengthening the biochemical barriers of the gut and intestine [11,12,13,14]. In many species, major shifts in the gut microbiota composition have been observed with IBD (e.g., [9,15]), although it remains unanswered whether gut microbiota dysbiosis is a cause or a consequence of IBD, or both.

As in humans and other animals, the prevalence of IBD in horses has dramatically increased during the past few decades [16]. Equine IBD is difficult to diagnose because the GIT of an adult horse is 30 m long. Endoscopy can be typically performed only on the uppermost (i.e., duodenum) or lowermost (i.e., rectum and distal colon) part of the GIT, leaving a substantial part of the GIT beyond reach. Additionally, the interpretation of endoscopic biopsies can pose challenges to pathologists [17]. To tackle challenges related to the diagnosis of equine IBD, developing a non-invasive, affordable, and reliable method to detect it is an important step. We propose to start this exhaustive task by identifying the characteristic features of the gut microbiota of horses with IBD from fecal samples. This method is affordable and not stressful for the horse as sampling is non-invasive and does not require fasting or transportation.

Fecal samples have been utilized in studies focusing on alterations in gut microbiota composition in horses with colitis, antimicrobial-associated diarrhea, and equine metabolic syndrome [18,19,20]. IBD has been shown to alter gut microbiota composition in other domestic animals, such as dogs and cats [9,15]. In humans, gut microbiota composition can be used to reliably predict the presence of IBD, to distinguish between different types of IBD, and to differentiate the state of the disease [21,22,23]. Interestingly, the association between gut microbiota composition and IBD has not yet been studied in horses. Here, we investigate for the first time the potential for IBD to alter gut microbiota composition in horses. In future, a comprehensive characterization of the horse’s gut microbiota can create new opportunities for early diagnostics and innovative therapeutic approaches for disease prevention and treatment [24].

In human medicine, supportive diagnostics for IBD have also been developed based on fecal calprotectin concentration. Calprotectin is a cytosolic protein primarily derived from neutrophilic cells and its presence in feces increases as a consequence of intestinal inflammation [25,26]. In humans, calprotectin concentrations have also been used in monitoring IBD activity and the response to treatment [27]. Calprotectin concentration has also been used as a biomarker for IBD or intestinal inflammation in other animals, such as dogs and cats [28,29]. In horses, changes in calprotectin concentration in epithelial cells have been associated with conditions such as laminitis [30]. However, few studies have utilized fecal samples and no studies have attempted to use it as a biomarker for IBD. If fecal calprotectin concentration proves to be a reliable biomarker for IBD, it could further contribute to the future development of a non-invasive diagnostic technique for equine IBD.

Here, we use fecal samples to study differences in hindgut microbiota composition and calprotectin concentration between healthy horses and horses with IBD. Based on the microbiota composition, we construct an artificial neural network model capable of predicting the presence of IBD at the individual level. We ask whether (1) IBD alters gut microbiota composition in horses, (2) certain bacteria taxa extracted from fecal samples and/or calprotectin concentration can be used as a diagnostic biomarker for IBD in horses, and (3) an artificial neural network model can be used to predict IBD in individual horses based on microbiota composition from fecal samples. Our results can be used to develop a non-invasive, relatively inexpensive diagnostic method for IBD in horses.

## 2. Materials and Methods

### 2.1. Sample Collection and Background Information on Horses with IBD

Fecal samples were collected before grazing season in 2023 to minimize the effect of rapid changes in diet on gut microbiota composition during that period. Sampling was conducted from fresh fecal samples, which were stored frozen until DNA extraction. We obtained case control sample pairs from 27 healthy horses and 30 horses with IBD located in the same stables from volunteers recruited from a social media campaign. Those samples had variable IBD onsets and activity statuses and are referred to as “survey IBD samples” from now on. A second set of 19 samples of IBD horses was obtained from Laukaa Horse Hospital (Laukaa, Finland). These are referred to as “acute IBD samples”. Horse owners were asked to fill in a questionnaire with questions about, for example, the horse’s age, gender, body condition, and symptoms, as well as other possible health issues the horse might have. We used a five-step body condition scale to create a body condition score (Appendix A).

We required that the control horses lived at the same stable as the survey IBD-horse pair, were not younger than two years old, and did not have any major known health issues. Among control horses, 48% were mares and among IBD-horses 41% were mares (Appendix A). The average age of control horses was 11.0 years (SD = 5.1) and IBD-horses 9.0 years (SD = 4.7; Appendix A). The average age for diagnosis was 8.2 years (SD = 4.7). The average body condition score of the control horses was 3.2 (SD = 0.7) and of IBD-horses 3.3 (SD = 0.4; Appendix A), which is qualified as “normal”. The control and IBD-horses did not differ significantly in terms of gender (χ2=0.174, *p* = 0.917), age (t = 1.610, *p* = 0.115), or body condition score (t = -0.302, *p* = 0.764). In 43.6% of the cases, IBD was diagnosed only by rectal biopsy and in 7.8% by small intestine biopsy (from the foregut). Additionally, 35.9% was diagnosed only by ultrasound, 5.1% only by gastroscopy and 5.2% by a combination of methods (e.g., gastroscopy and ultrasound). Of the IBD-diagnoses, 28% were eosinophilic enterocolitis, 23% lympho-plasmasytic colitis/proctitis, and the rest were rarer or non-defined subtypes. Of the horses diagnosed with IBD, 39% were mares, 57% were geldings, and 4% stallions (in the data analyses geldings and stallions were combined). Based on the questionnaire the horse owners filled in (N = 29 of total 31 survey IBD-horse owners, N = 10 of total acute IBD-horse owners, and N = 20 of total 26 control horse owners), 26% of the horses were suffering from an acute IBD and the rest were in remission. The majority of IBD-horses were of normal body condition (61.5%), 25.6% were fat, and 5.1% were very fat. Only 7.7% of the IBD-horses were thin and none of the IBD-horses were very thin. The questionnaire revealed that 75.6% of the owners reported serious or moderate pain-related behaviors in their IBD-horses, 62.8% reported seriously or moderately lowered performance, 53.7% reported serious or moderate aggressive behavior, and 29.3% reported that the horse had serious of moderate problems with riding. While 40% of the horses with IBD had also other health issues, such as gastric ulcer or cushings, only 28% of the control horses had health problems. Among IBD-horses, 54.4% had caries (compared to 33% of the control horses) and 20% had Equine Odontoclastic Tooth Resorption and Hypercementosis (EOTRH; compared to 0% of the control horses). For the 17 horses for which the questionnaire was not filled, we were able to fill age and gender based on Heppa database of The Finnish Central Organisation for Trotting and Horse Breeding (Hippos), except for three horses.

### 2.2. DNA Extraction and Sequencing 

DNA extraction was conducted using the DNeasy PowerSoil Pro Kit and the associated protocol with the Qiagen TissueLyser II cell disruption step (Venlo, The Netherlands). Libraries were prepared at the University of Jyväskylä using Illumina’s MiSeq 16S amplicon sequencing protocol. Amplicons targeted the V3–V4 variable regions [31] of the 16S ribosomal RNA (rRNA) locus, which were amplified by a polymerase chain reaction using the 341F (5′-TCGTCGGCAGCGTCAGATGTGTATAAGAGACAGCCTACGGGNGGCWGCAG-3′) and 806R (5′-GTCTCGTGGGCTCGGAGATGTGTATAAGAGACAGGACTACHVGGGTATCTAATCC-3′) primer sets [31,32]. Libraries were sequenced on an Illumina MiSeq2000 (San Diego, CA, USA) to provide 250 base paired-end (PE) reads, with the aim of obtaining 100,000 read pairs per sample.

### 2.3. Sequence Processing

We merged read pairs for each sample using FLASH v.1.2.11 software with a read length of 250 bases and an insert size of 430 bases [33]. FLASH merged 49 639–156 944 read pairs (89–96% of the raw reads). Then, we assembled operational taxonomic units (OTUs), assigned the merged reads between OTUs, and inferred the taxonomy of each OTU using the Qiime2 v. 2023.9 amplicon pipeline [34]. Briefly, we used vsearch to dereplicate the read sequences, followed by vsearch open-reference clustering, which combined reference-based clustering with Silva reference sequences v. 13.899 and de novo clustering to assemble OTUs. In the clustering step, 98% similarity was used. A taxonomic 16S V3-V4 classifier was built from the Silva reference sequences v. 13.899, which were filtered by removing low-quality sequences and non-eukaryotic sequences. The 16S V3-V4 region was then extracted using primers 5’-CCTACGGGNGGCWGCAG-3’ and 5’-GACTACHVGGGTATCTAATCC-3’, and the final classifier sequences were dereplicated. Taxonomy annotations of the equine OTUs were inferred from the classifier using the sklearn machine learning classification approach. We filtered the OTU counts by requiring that at least 10 reads were detected in two samples at the taxonomic unit used. To generate a count set that was normalized based on the sequenced read counts, we used the rrarefy function in the vegan v. 2.6.4 package and rarefied the observed counts of each individual to the smallest observed total count per individual.

Sequencing resulted in 54,389–165,782 read pairs in each sampled individual. The pipeline resulted in 6,863,691 merged reads associated with 3.1 million OTUs. In each OTU, we detected 1–28,081 merged reads across all samples. The total number of merged reads per sample ranged from 49,639 to 156,944. After removing rare OTUs, 37,281 OTUs and a total of 3.3 million merged reads (47.5% of all merged reads) remained (see Appendix A for assembly statistics). The counts were rarefied to the smallest sample, which was 23,335 merged reads per individual (Appendix A).

### 2.4. Calprotectin Extraction

We used the Horse Calprotectin ELISA kit (MBS038554; enzyme-linked immunosorbent assay; MyBioSource, San Diego, CA, USA) to determine the calprotectin concentration of fecal samples. For the determination of calprotectin concentration, we used fecal samples from 14 horses in the acute IBD group and from 14 healthy horses. We collected 0.5 g of feces per sample in an Eppendorf tube and added 2500 µL of PBS buffer solution into the tube. Samples were vortexed and placed in the freezer for 24 h before the analysis, which was performed according to the instructions included in the Horse Calprotectin ELISA kit. The kit included standard solutions to estimate a standard curve, which was used to estimate the calprotectin concentration of the fecal samples.

### 2.5. Statistical Analyses of the Sequence Data

#### 2.5.1. Alpha- and Beta-Diversity

We estimated microbial species richness within individuals (alpha-diversity) from the rarefied OTU counts. For this, we used the estimateR and diversity functions in vegan package v. 2.6.4 to calculate Chao1, evenness, and Shannon metrics. We used four-factor ANOVA to assess whether the rank-scaled alpha-diversity estimates differed between the three groups of horses (i.e., control, survey IBD samples, and acute IBD samples), also accounting for variation explained by age, gender (mare or gelding/stallion), and reported body condition score. Missing observations were imputed using medians.

Beta-diversity describes the differences in microbiota composition between individuals. To quantify the strength of the association between microbiota composition (beta-diversity) and health status, we performed a distance-based redundancy analysis implemented in the capscale function of the vegan v. 2.6.4 package, using 9 999 permutations and Jaccard distance in the analysis. Then, we assessed whether the two IBD groups (i.e., survey and acute) were consistently different from the control group among OTUs which were different between study groups. We used *t*-tests to identify candidate OTUs with un-adjusted *p* < 0.05 in either control vs. survey IBD or control vs. acute IBD comparisons. We calculated the differences between mean OTU abundances in control samples and those in survey IBD samples and repeated this for acute IBD samples. We then assessed the consistency between the survey IBD and acute IBD differences using Pearson’s correlation. To exclude bias, which would have been caused by using the same control individuals in both comparisons, we divided the control horses into two randomly assigned groups. We bootstrapped the correlation 100 times to assess the effect of control horse selection. Then we used a *t*-test to compare the bootstrapped correlation distribution to an empirical null distribution of correlations using 100 permutations of random labeling.

The effect of health status, age, gender, and body condition on log10-transformed Firmicutes/Bacteroidota (F/B) abundance ratios was tested using ANOVA and Pearson’s correlation when explanatory factors significantly explained the F/B ratio. 

#### 2.5.2. Differentially Abundant OTUs

To test differentially abundant OTUs between the control and IBD groups, we used ANCOMBC2 v.2.2.2 [35]. We used the health status (control, survey IBD, or acute IBD), age, gender (mare or gelding/stallion), and body condition score (Appendix A) as fixed variables. Missing observations were imputed using medians and *p*-values were corrected according to the Benjamini–Hochberg FDR procedure for multiple testing. We extracted differentially abundant OTUs using an adjusted significance level of *FDR* < 0.05 and inspected whether they passed the sensitivity filter despite having the pseudo count in the model. We defined the OTUs that were assigned as differentially abundant in both comparisons (control vs. survey IBD and control vs. acute IBD) as consistently differentially abundant if the log2-fold changes were either positive or negative in both IBD groups. We used the chi-squared test to assess whether the consistent abundancy changes were more frequent in the differentially abundant OTUs than in the non-differentially abundant OTUs. We also used the chi-squared test to assess whether the Firmicutes/Bacteroidota (F/B) species count ratio among the differentially abundant OTUs from the IBD comparisons deviated from the overall count ratio over all taxa.

#### 2.5.3. Artificial Neural Network Model

To test the power of fecal samples as diagnostic markers, we used a shallow feed-forward artificial neural network model to predict control or IBD status from the 200 most informative OTU counts, inferred using Wilcoxon signed-rank tests with *p* < 0.05. Prior to modeling, each OTU count was scaled and centered. Then, 80% of the samples were assigned to a training group and 20% of the samples were assigned to a test group. The training group was used with the health status labels (i.e., IBD or healthy) to teach the model to recognize patterns in OTU composition, which predicted whether the horse had IBD or not. We merged the two IBD groups (survey and acute) and used sample weights to balance the model for unequal group sizes. We used the logistic activation function and tested the size of three, six, and nine nodes for the hidden layer, as overfitting became an issue with larger sizes, and tested decay parameters of 0.1, 0.2, 0.3, and 0.4. We validated the model with a repeated *k*-fold cross-validation procedure using five repeats and five folds. After training the model, we validated its performance (specificity and sensitivity) by predicting the status of the test set of samples, which were previously unseen by the model. R packages caret v. 6.0.94 and nnet v. 7.3.19 [36,37] were used in the diagnostic model development step.

## 3. Results

### 3.1. Microbiota Composition

Our samples revealed a typical range of taxa associated with equine fecal microbiota [38]. The core microbiota was estimated from healthy, control samples. The microbiota was dominated by phyla Firmicutes (on average 48.7% of the rarefied reads) and Bacteroidota (22.5%). In addition, phyla Verrucomicrobiota (12.3%), Spirochaetota (4.2%), Fibrobacterota (3.1%), and Actinobacteriota (1.2%) were relatively abundant, while each of the rest of the 17 detected phyla constituted less than 1% of the total reads (Appendix A, Appendix A). Many of the most abundant families (e.g., Lachnospiraceae; on average 16.6% of reads) belonged to the most dominant phyla (e.g., Firmicutes; Appendix A). However, some of the most common families (such as WCHB-41, which made up 12.0% of all bacteria) belonged to less common phyla (such as Verrucomicrobiota; Appendix A, Appendix A). The Firmicutes/Bacteroidota (F/B) ratio was 1.88.

### 3.2. Differences in Microbiota Composition between Healthy Horses and Horses with IBD

There were no differences in alpha-diversity (diversity within individual species) between the groups (control, survey IBD samples, and acute IBD samples) based on Chao1 (ANOVA, F_2,70_ = 2.064, *p* = 0.135), evenness (F_2,70_ = 2.750, *p* = 0.445), and Shannon diversity index (F_2,70_ = 3.120, *p* = 0.050; Appendix A). Age, gender, or body condition did not explain differences in alpha-diversity (Appendix A). However, there was an association between microbiota composition (beta-diversity), health status (control, survey IBD samples, and acute IBD samples; ANOVA, F = 1.142, *p* = 0.030), and the age of the horses (F = 1.306, *p* = 0.011; Figure 1). CAP1 separated the control and acute IBD horses (F = 1.515, *p* = 0.005), whereas CAP2 separated the control and survey IBD horses (F = 1.041, *p* = 0.616). Together, age and health status explained 4.7% of the differences among groups. Gender and body condition did not have a statistically significant effect on the microbiota composition difference among groups. The suggestively differentially abundant OTUs were more consistently changed between control and survey IBD and control and acute IBD comparisons (*r* = 0.424) than expected by null distribution (t_183.73_ = −12.87, *p* < 0.001).

The F/B ratio did not differ among control, survey IBD, and acute IBD samples (ANOVA, F_2,70_ = 0.003, *p* = 0.910; Appendix A, Appendix A). No variance was associated with gender or body condition score, but age was a significant explanatory variable (F = 6.480, *p* = 0.010; Appendix A), which correlated negatively with the F/B ratio (r = −0.28, t_74_ = −2.52, *p* = 0.014).

### 3.3. Differentially Abundant OTUs

We detected 11,294, 3465 and 368 differentially abundant OTUs, which were sensitive to age, gender, and body condition score, respectively (*FDR* < 0.05). Out of those, 486,175 and 13 OTUs passed the sensitivity filter of ANCOMB2, suggesting that the results may be sensitive to whether a pseudocount is added to the model and should be further inspected. A total of 11 and 41 differentially abundant OTUs were detected in control vs. survey IBD and control vs. acute IBD group comparisons, respectively (Figure 2, Appendix A). Out of these, only three from both comparisons passed the sensitivity filter. As the majority of the differentially abundant OTUs did not pass the sensitivity filter, more evidence on these differentially abundant OTUs was required. Thus, we verified the relevance of this observation by comparing the fold changes of OTU abundances between the control group and the two IBD groups. Among OTUs, which were differentially abundant in at least one of the comparison fold changes, these were consistent (OTU abundance either increased or decreased both between control vs. survey IBD groups and between control vs. acute IBD groups) in 85.4% of the cases. This was more than among the non-differentially abundant OTUs (χ^2^ = 7.800, *p* = 0.005).

The F/B ratio of the observed number of species decreased from 2.7 among all OTUs to 0.6 among differentially abundant OTUs in the control vs. survey IBD comparison (χ^2^ = 12.32, *p* < 0.001) and to 0.7 (χ^2^ = 20.78, *p* < 0.001) in the control vs. acute IBD comparison. The abundancy of Firmicutes in OTUs, which were differentially abundant in one of the control vs. IBD comparisons, was 17.1% of the observed reads.

### 3.4. Prediction of IBD Prevalence with the Artificial Neural Network Model

Using the artificial neural network model in IBD status prediction, the best prediction accuracy was obtained using nine hidden nodes and a decay of 0.3. Cross-validation resulted in up to 96.8% accuracy (SD = 5.2%) and a Kappa value of 93.4% (SD = 10.5%). The importance distribution of input nodes (OTUs) is shown in Appendix A and Appendix A. The remaining test set (20% of the samples with six control samples and 10 IBD samples) was correctly predicted by the model, thus resulting in 100% sensitivity and 100% specificity (Figure 3).

### 3.5. Fecal Calprotectin Measurement

We did not detect differences in the average calprotectin concentration between healthy and IBD-affected horses (t_25.43_ = 1.217, *p* = 0.235; Appendix A). The range of fecal calprotectin concentration was 57.3 ± 9.48 µg/g in IBD horses and 50.6 ± 13.09 µg/g in healthy horses.

## 4. Discussion

We showed that the overall microbiota composition of horses with IBD differed from that of healthy horses. This difference was particularly clear between healthy horses and horses with acute IBD (i.e., IBD that was diagnosed at the Laukaa horse hospital, where the samples were taken). We further identified a number of OTUs that were associated with equine IBD. While bacterial composition could be used as a potential biomarker in IBD diagnosis (particularly when occurring in the hindgut), fecal calprotectin concentration proved to be unsuitable. We trained and used an artificial neural network model to predict the probability of IBD. While we acknowledge that more samples are needed in future with different parameters, such as sampling batch and more defined IBD subtypes, to improve the usability of the model, it predicted the probability of IBD with 100% sensitivity and specificity in our relatively small data set.

Contrary to our initial expectations, alpha-diversity was not affected by the presence of IBD, at least not at the level of resolution provided by 16S rRNA gene sequencing. Human studies have shown decreased alpha-diversity in patients with IBD (e.g., [39,40]) and studies with rodent models have documented similar decreases in alpha-diversity (e.g., [41]). Changes in gut microbiota diversity (or composition) associated with IBD have thus far not been studied in horses. However, in various studies comparing horses with acute colitis with healthy controls, no consistent evidence of decreasing alpha-diversity was found (reviewed in [38]). Similarly, inconsistent results have been reported for cows affected by Johne’s disease, which resembles human IBD [42,43,44]. It can be speculated that alpha-diversity in large grazers with a long gastrointestinal tract (GIT), such as cows and horses, is less affected than in animals with a relatively short GIT, such as humans, because of differences in digestion rate (e.g., horses digest 10 kg of hay per day in their hindgut). In addition, some human studies report no significant difference in alpha-diversity between healthy individuals and individuals suffering from ulcerative colitis [45], whereas [46] showed that alpha-diversity decreased in human patients with acute severe colitis. However, this decrease was less prominent in patients with mildly to moderately active ulcerative colitis. Indeed, our samples consisted of individuals with various IBD activity stages (and subtypes), which could have affected the results. However, we found no difference in alpha-diversity even between acute IBD samples (collected at the horse hospital) and healthy, control samples. Considering the non-uniformity of the diagnostic methods that are currently being applied (evident in our survey IBD samples), the variation associated with our relatively heterogenic study group may partly explain the lack of differences in alpha-diversity between horses with IBD and healthy, control horses.

Despite no differences in alpha-diversity, we found an association between microbiota composition and IBD using beta-diversity analysis. This result is in line with numerous IBD studies in humans (reviewed in [47]), model organisms (e.g., [48,49]), and domestic animals (reviewed in [4]). The gut microbiota composition of the healthy, control horses and acute IBD horses (samples collected at the horse hospital) differed significantly from each other (*p* = 0.005), while the control group and survey IBD samples did not. This is understandable, as the survey IBD samples were relatively heterogenic in terms of, for example, IBD stage. As mentioned earlier, in approximately a quarter of the surveyed IBD horses, IBD was at an acute stage and was in remission in the rest of the horses. Thus, the gut microbiota composition of recovering individuals may resemble that of the healthy, control individuals. Indeed, a study on horses with diarrhea showed that their microbiota composition started to look less different than that of healthy horses after they received treatment (fecal microbial transplantation [20]). Despite health status (and age) having a significant effect on gut microbiota composition, only less than five percent of the total variation in microbiota composition was explained by these variables. This could be due to two reasons. First, the survey IBD samples were collected from horses with variable IBD activity stages and some of them had already received treatment. It is also good to keep in mind that only survey IBD horses had a healthy control horse pair sampled from the same stable. Therefore, one may conclude that the differences between control horses’ and acute IBD horses’ gut microbiota composition could be caused by the differences in living environment, feeding, and husbandry. We cannot entirely exclude the possibility that the difference between the control and acute IBD samples reflects differences in management, as those samples were collected from different stables. However, we did observe that OTU abundances were more likely to be consistent between the comparisons of the control and two IBD groups (survey and acute) than expected by random. This supports the conclusion that the differences were caused by IBD and not entirely by different environments. Second, IBD is not a single disease but describes disorders of idiopathic, chronic, and relapsing conditions in the GIT, which are relatively poorly understood and defined in horses. This could further contribute to the low percentage of variation in microbiota composition explained by IBD. While the exact cause of IBD is still unknown, a weakened immune system, environmental triggers, genetic background, and pathogens are known to be associated with IBD [50,51,52]. In addition, oxidative stressors which boost reactive oxygen species generation and antioxidant hormones have been suggested to play a role in the etiology of IBD [53]. Gut microbiota dysbiosis, which has been described rather ambiguously as “alterations in gut microbiota composition and function”, has also been associated with IBD by numerous studies (e.g., [54,55,56]). However, based on our results, we would be rather careful concluding that IBD might have caused a microbiota dysbiosis in the hindgut of IBD-diagnosed horses.

We do acknowledge that when using gut microbiota compositions obtained from fecal samples, a better representation of the microbiota composition in the hindgut (i.e., the large intestine, which includes the cecum, large colon, small colon, and rectum) is provided compared to the foregut (i.e., the small intestine, which includes the duodenum, jejunum, and ileum) [57]. Despite the microbiota composition being known to differ in different parts of the GIT, it does not necessarily mean that fecal samples cannot be used in detecting IBD occurring in the small intestine. In fact, human studies have demonstrated that fecal samples can be used in detecting biomarkers for IBD (Crohn’s disease) diagnosis in the small intestine [21,58,59]. In horses, future studies should focus on comparing microbiota composition in fecal samples between horses with IBD in the foregut and in the hindgut and assessing whether the location of IBD can be detected using fecal samples. We further highlight the importance of studying different IBD subtypes, whether gut microbiota composition differs among different subtypes, and how they vary among global populations and in response to management and potentially diagnostic methods. These studies would benefit from using well-defined and thoroughly diagnosed IBD subtypes from euthanized individuals.

Everyone’s gut microbiota composition is unique, but several studies across species have identified that each species has a similar core of bacteria, known as the core microbiota [60,61]. The core microbiota refers to the microbiota that occurs and originates from similar habitats and plays a significant role in maintaining the stability, plasticity, and function of the gut, remaining relatively stable regardless of time and place [62]. Our results were consistent with previous studies showing that the horse core microbiota consisted mostly of Firmicutes, Bacteroidota, Verrucomicrobiota, and Spirochaetota (e.g., [24,63]). The Firmicutes to Bacteroidota ratio (F/B ratio) has been suggested to be a relevant marker of gut microbiota dysbiosis [64] and that this ratio increases in horses suffering, for example, from intestinal colic compared to healthy controls [65]. Interestingly, we found no difference in the F/B ratio between healthy horses and horses with IBD. The abundance of Firmicutes has been shown to respond (either increase or decrease) to frequently administered medications [66,67] and diarrhea [68,69]; however, we show that only 17% of differentially abundant OTUs in acute IBD horses were (otherwise dominant) Firmicutes. This means that more than ¾ of the differentially abundant OTUs belonged to other phyla than Firmicutes in acute IBD horses, even though Firmicutes is the most abundant phyla in the gut microbiota.

We detected approximately 40 differentially abundant OTUs associated with IBD. The majority of these were consistently affected in both survey and acute IBD groups. This suggests that the observed differences may have biological relevance. With the ANCOMB2 approach, we identified significantly fewer differentially abundant OTUs. Identifying single taxa with differential abundance between healthy and IBD horses was challenging, likely due to power issues stemming from the inherent complexity of microbial data. This complexity may arise from zero-inflated counts and interactions among the microbial taxa. Furthermore, problems may arise because of the relatively low sample size per study group (although our sample size was not unusually low compared to previous horse gut microbiota studies) and the low abundance of differentially abundant taxa. In future, identifying bacteria at strain level by utilizing longer reads may facilitate the identification of causal taxa with sufficient resolution. Alternatively, the changes in microbial composition due to chronic diseases, such as IBD, may cause relatively subtle changes in the microbial ecosystem and these may be hard to detect using isolated taxa.

We successfully detected IBD from the fecal samples using the microbial community profiles as input in our machine learning approach. Despite moderate sample size and heterogenic data (the survey IBD samples), the diagnostic power of the combined microbial profiles far exceeded the methods for IBD detection that are currently being utilized. Importantly, we used a subset of test samples to obtain an unbiased estimate of the final model’s performance with novel samples, which were not used in model construction. We acknowledge that differences in sequencing methods, such as the primers used, library preparation protocols, or the sequencing device used can lead to different results, although it has been shown that, for example, the core gut microbiota of horses does not differ regardless of whether the composition is inferred using amplicon sequencing or metagenomic sequencing [70]. As the next step, the usability of our model should be improved by training the model with stochastically varying data, such as samples from different sequencing batches and time points (temporal variation). We further aim to train the model to distinguish different IBD activity stages and differences between IBD and other gut health-related problems.

## 5. Conclusions

Current IBD diagnostics in horses rely on invasive, stressful, and expensive methods such as intestinal biopsies. The development of non-invasive techniques, such as GIT microbial composition analysis for biomarker discovery, could improve equine IBD screening. Based on our data, the microbiota composition obtained by 16S sequencing fecal samples differs between IBD and healthy horses and could therefore become part of the diagnostic work-up. Fecal calprotectin concentration did not seem to be of added benefit in IBD diagnostics in horses. Thus, we conclude that utilizing microbial communities of fecal samples is a promising avenue for cost-efficient screening of IBD in horses.

## Figures and Tables

**Figure 1 animals-14-02396-f001:**
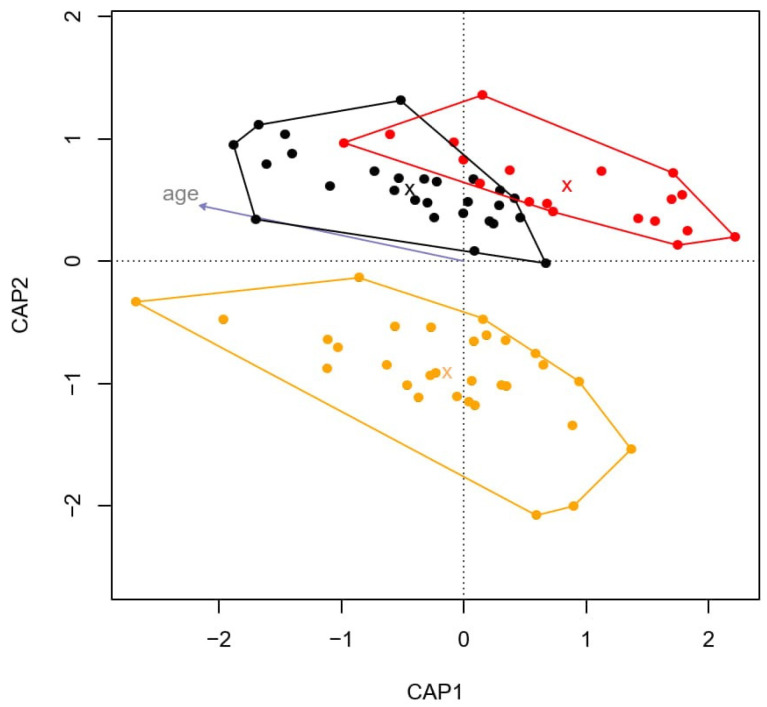
Distance-based redundancy analysis grouped healthy and IBD horses into distinct clusters. Black circles represent samples of healthy horses, orange circles represent survey IBD horses, and red circles represent acute IBD horses. Group averages are marked with X and the clusters are defined based on the outermost values among each study group. Additionally, the individual eigenvalues of the individuals are correlated with the age of the horse (gray arrow).

**Figure 2 animals-14-02396-f002:**
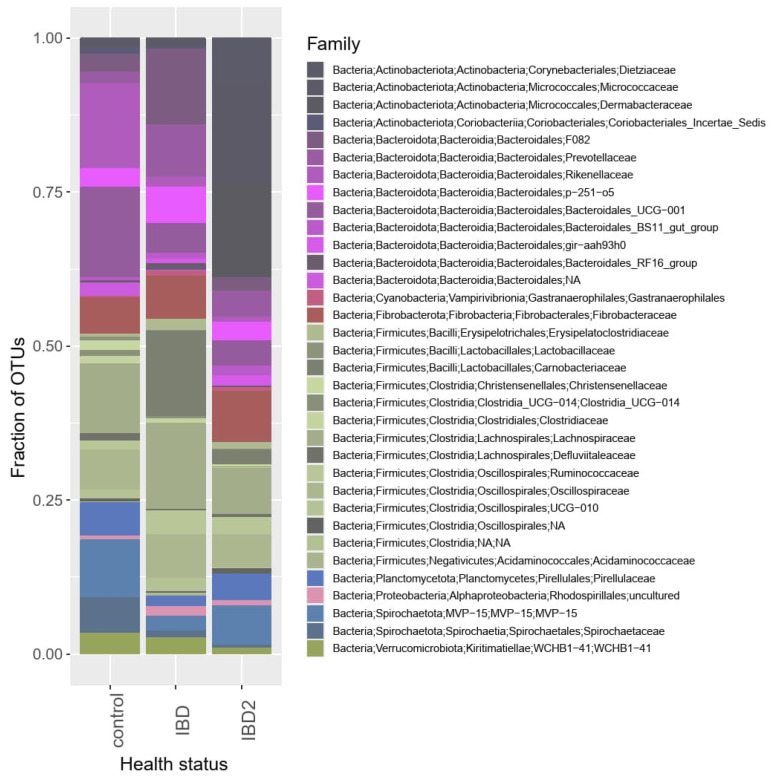
Family-level relative abundancies of control, survey IBD, and acute IBD horses summed together among the rarefied counts of differentially abundant taxa. A total of 24 and 69 differentially abundant OTUs from control vs. survey IBD and control vs. acute IBD comparisons, respectively, with *FDR* < 0.1 were selected to be presented in this figure as they had almost as many consistently abundant OTUs (82.4%) as differentially abundant OTUs with a correction of *FDR* < 0.05. Complementary Figure with all taxa regardless of the differentiation level can be found in the Appendix A.

**Figure 3 animals-14-02396-f003:**
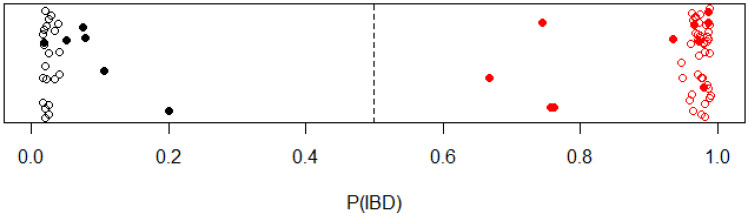
The probability of having IBD for each sample from the artificial neural network model. Control and IBD samples (survey and acute) are indicated in black and red, respectively. The 80% of samples used in the cross-validation are indicated with open circles. The post-modeling test group (20% of samples) are marked with filled circles.

## Data Availability

The data that support the findings will be available in Jyväskylä University Digital Repository at www.jyx.jyu.fi (accessed on 7 August 2024) following an embargo from the date of publication to allow for commercialization of research findings.

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
