# Peer review of "Gut Microbiota Profiling as a Promising Tool to Detect Equine Inflammatory Bowel Disease (IBD)"

_animals, 2024, doi:10.3390/ani14162396_

Round 1

Reviewer 1 Report

Comments and Suggestions for Authors

The authors of this article showed that horses with inflammatory bowel disease have differences in their faecal microbiota compared to healthy horses with the potential of becoming a non-invasive diagnostic tool in diagnosing IBD in horses. This field or research is very interesting, and this article is relevant for equine veterinary field. 

Article review points:

Materials and Methods:

-       Unclear how the used IBD horses were diagnosed: Duodenal biopsies? Rectal biopsies? Glucose absorption test, rectal thickened intestinal walls etc?

-       Unclear where the survey horses where in their disease: in remission or continuously treated etc. (mentioned in the discussion) and how this would affect the results, maybe discuss further?

Discussion:

-       Survey IBD horses were not different from the controls, but the survey IBD horses and the controls came from the same stables. In the discussion this was attributed to the heterogenicity of the IBD-status of the IBD survey group.

The acute IBD cases did not come from the same stables as the controls, and they were different. Could the different feed and management of this group not be a cause of this difference versus the IBD? Should there have been controls from the same stables for the acute IBD group in the control group as well, maybe discuss further? 

-       Technical differences in the methods of microbiota sequencing can provide very different results so this should be discussed.

-       Microbiota is very different from foregut and hindgut in horses, IBD is mainly focussed in the foregut in horses so faecal samples might not be the most reliable, could this not have affected the results?

-       IBD in general is more located in the foregut, although hindgut can be involved. Percentages of IBD sub-types used in this study seem to differ from literature. It seems a high percentage of eosinophilic enterocolitis and lymfo-plasmacytic colitis and even proctitis versus lymfo-plasmacytic enteritis which is generally seen the most. Both subtypes include colitis and even proctitis which might give very different results in the faecal samples used versus the more cranial located IBD, maybe discuss further? 

-       Key question about cause or consequence in introduction is not clearly mentioned again in discussion in R 372, maybe discuss a bit further as this is quite interesting.

Conclusion:

-       Conclusion could be more concise: 

investigating microbiota in fecal samples in IBD horses by 16S sequencing seems to be different than healthy horses on …….  and could therefore become part of the diagnostic work-up. 

Calprotectin does not seem to be of added benefit in the IBD diagnostic work-up in horses.

Specific comments:

R 71      More recent publications are available and clear treatments are currently proposed including prognostic indicators beyond “anecdotal reports of success”.

R 73      The test currently used in diagnosing IBD is the “glucose (oral) absorption test”. The “oral glucose tolerant test” is used for diagnosing EMS, or do the authors mean something else?

The glucose (oral) absorption test (in diagnosing IBD) does require fasting and a nasogastric tubing to administer the glucose and the taking of repeated blood samples and is therefore not considered non-invasive.

R 231    34 IBD horse owners filled in a questionnaire and were considered acute IBD. We had 30 IBD survey horses and 19 acute IBD cases, so were there only 34 questionnaires filled in and from which group were they?

R 369    numerous studies, please provide references

             R 425    “not used in training” meaning is unclear. 

Author Response

REVIEWER 1 

The authors of this article showed that horses with inflammatory bowel disease have differences in their faecal microbiota compared to healthy horses with the potential of becoming a non-invasive diagnostic tool in diagnosing IBD in horses. This field or research is very interesting, and this article is relevant for equine veterinary field.   

Materials and Methods 

Unclear how the used IBD horses were diagnosed: Duodenal biopsies? Rectal biopsies? Glucose absorption test, rectal thickened intestinal walls etc? 

RESPONSE: 43% of the IBD-horses were diagnosed with rectal biopsies and 8% with duodenal biopsies. Glucose absorption test was not used in diagnosis. 36% were diagnosed with ultrasound (thickened intestinal walls). All acute IBD-horses (i.e., samples received from the Laukaa horse hospital) were diagnosed with rectal biopsies. We have now added this information in the revised Material and Methods (page 3, lines 113-117). 

Unclear where the survey horses where in their disease: in remission or continuously treated etc. (mentioned in the discussion) and how this would affect the results, maybe discuss further? 

RESPONSE: We have now added this information in the revised Material and Methods (page 3, lines 123-124) and discussed it in the revised Discussion (page 10, lines 389-394). 

Discussion 

Survey IBD horses were not different from the controls, but the survey IBD horses and the controls came from the same stables. In the discussion this was attributed to the heterogenicity of the IBD-status of the IBD survey group. The acute IBD cases did not come from the same stables as the controls, and they were different. Could the different feed and management of this group not be a cause of this difference versus the IBD? Should there have been controls from the same stables for the acute IBD group in the control group as well, maybe discuss further?  

RESPONSE: Thank you for pointing this out. To confirm that the differences between control and acute IBD-horses were not solely due to living conditions, feeding, and husbandry we have now conducted an additional analysis showing that the two IBD groups (survey and acute) were consistently different from the control group. We have explained this analysis in revised Material and Methods (page 5, lines 201-212) and added the results in the revised Results (page 6, lines 277-279). These results are also shown in the figure below and discussed in the revised Discussion (page 10, lines 398-407).

The OTU abundance changes between control samples and survey-IBD group were consistent (correlated positively) with OTU abundance changes between control and acute-IBD samples. The positive correlation is presented as a distribution (red histogram) of correlations obtained by 100x bootstrapped selection of control samples, which we divided randomly to two groups, one for each comparison. This was necessary to exclude bias caused by non-independence of the control samples if they would be included in both comparisons. The gray distribution is empirical null, which we obtained from 100x permutations of sample labels.

Technical differences in the methods of microbiota sequencing can provide very different results so this should be discussed. 

RESPONSE: In our study, all samples were sequenced with the same method (library prep and sequencing device) and in the same sequencing batch. However, the reviewer is correct: different protocols in library preparation, using different sequencing devices or methods, such as metagenomic sequencing can lead to different results. We have now mentioned this in the revised Discussion (page 11, lines 468-472). 

Microbiota is very different from foregut and hindgut in horses, IBD is mainly focussed in the foregut in horses so faecal samples might not be the most reliable, could this not have affected the results? IBD in general is more located in the foregut, although hindgut can be involved. Percentages of IBD sub-types used in this study seem to differ from literature. It seems a high percentage of eosinophilic enterocolitis and lymfo-plasmacytic colitis and even proctitis versus lymfo-plasmacytic enteritis which is generally seen the most. Both subtypes include colitis and even proctitis which might give very different results in the faecal samples used versus the more cranial located IBD, maybe discuss further?

RESPONSE: We realize that fecal samples do not well represent the microbiota in the foregut and therefore potentially cannot help in diagnosis when IBD occurs there. However, our study was not intended to describe the differences in microbiota composition in different parts of the GIT but to find an association between hindgut microbiota composition and IBD. Admittedly, we only had quite few horses with IBD diagnosed in the foregut but those still clustered together with horses with IBD diagnosed in the hindgut. Furthermore, human studies indicate that fecal samples can be used in diagnosing IBD (Chron’s disease) in the small intestine (Tedjo et al. 2016; Hattori et al. 2020; Olbjørn et al. 2022). We acknowledge that more research is needed to study whether fecal samples can detect IBD occurring in the foregut in horses. Reviewer #3 requested a significant shortening of the Introduction and we have now acknowledged this shortcoming and discussed it in the revised Discussion (page 10, lines 418-428). Furthermore, we have added IBD “particularly when occurring in the hindgut” in the revised Discussion (page 9, lines 352-353 and page 10, line 417). The IBD subtypes in our study are based on pathologist’s assessment and we acknowledge that further research is needed to investigate their occurrence in different horse populations in different countries and how they vary based on management and diagnostic methods (pages 10-11, lines 428-432). 

Key question about cause or consequence in introduction is not clearly mentioned again in discussion in R 372, maybe discuss a bit further as this is quite interesting. 

RESPONSE: Thank you for this comment, we have now discussed this briefly in the revised Discussion (page 10, lines 409-413). We chose not to speculate this extensively because 1) we deleted the sentence from the Introduction as the reviewer #3 requested to shorten the Introduction significantly and 2) our pilot study cannot provide an answer for this question at this point. 

Conclusions

Conclusion could be more concise:  

investigating microbiota in fecal samples in IBD horses by 16S sequencing seems to be different than healthy horses on …….  and could therefore become part of the diagnostic work-up.  

Calprotectin does not seem to be of added benefit in the IBD diagnostic work-up in horses. 

RESPONSE: We have now shortened the Conclusions according to the reviewer’s suggestions (page 12, lines 478-485). 

Specific comments 

R 71      More recent publications are available and clear treatments are currently proposed including prognostic indicators beyond “anecdotal reports of success”. 

RESPONSE: Reviewer #3 requested to shorten the Introduction, thus this part of the Introduction has been deleted. 

R 73      The test currently used in diagnosing IBD is the “glucose (oral) absorption test”. The “oral glucose tolerant test” is used for diagnosing EMS, or do the authors mean something else? The glucose (oral) absorption test (in diagnosing IBD) does require fasting and a nasogastric tubing to administer the glucose and the taking of repeated blood samples and is therefore not considered non-invasive. 

RESPONSE: Thank you for this comment. Interestingly, some papers (e.g., Boshuizen et al. 2018) refer to this test as non-invasive, although we agree that fasting, tubing and repeated blood samples are not what at least we consider non-invasive. We have now deleted this part of the sentence. 

R 231    34 IBD horse owners filled in a questionnaire and were considered acute IBD. We had 30 IBD survey horses and 19 acute IBD cases, so were there only 34 questionnaires filled in and from which group were they? 

RESPONSE: We apologize of being unclear about the number of horse owners who filled in the questionnaire. We have now clarified the sentence in the revised Material and Methods (page 3, lines 121-123 and lines 134-136). 

R 369    numerous studies, please provide references. 

RESPONSE: We have now added references from human and rodent model studies (page 10, line 415). 

R 425 “...not used in training” meaning is unclear.

RESPONSE: The sentence has been deleted as we shortened the Conclusions. 

Reviewer 2 Report

Comments and Suggestions for Authors

As a result of the need to search for a new non-invasive method of diagnosing IBD in horses, the authors of the article attempted to identify specific bacterial strains from fecal samples. In a further stage of the research, the obtained results were subjected to machine learning so as to obtain a diagnostic model of IBD. The work is clinically significant and needs only minor corrections.

Specific comments:

Line 51 - GIT instead of “gastrointestinal system”. Microbiota are not present in liver or pancreas which belong to gastrointestinal system.

Line 83 - the terms "foregut" and "hindgut" are not used in veterinary anatomy. Please introduce and use the appropriate anatomical names of horse GIT.

Line 133 - please describe how primares were designed and validated.

Line 171 - how many factor ANOVA and with what post-hoc test were used?

Line 225 - "background information of the IBD-horses" are not strictly results and should rather be moved to the description of the animals  (line 122).

Author Response

REVIEWER 2 

As a result of the need to search for a new non-invasive method of diagnosing IBD in horses, the authors of the article attempted to identify specific bacterial strains from fecal samples. In a further stage of the research, the obtained results were subjected to machine learning so as to obtain a diagnostic model of IBD. The work is clinically significant and needs only minor corrections. 

Specific comments 

Line 51 - GIT instead of “gastrointestinal system”. Microbiota are not present in liver or pancreas which belong to gastrointestinal system. 

RESPONSE: Thank you for this comment, the terminology has been changes accordingly (page 2, line 51). 

Line 83 - the terms "foregut" and "hindgut" are not used in veterinary anatomy. Please introduce and use the appropriate anatomical names of horse GIT. 

RESPONSE: These have been added in the revised Discussion (page 10, lines 419-421). 

 Line 133 - please describe how primers were designed and validated. 

RESPONSE: The gene‐specific sequences used in this protocol target the 16S V3 and V4 region. They were selected from the Klindworth et al. (2013) publication as the most promising bacterial primer pair. We have now added this reference and referred also to another study where the same primers were used to study horse gut microbiota composition (page 4, line 147).  

Line 171 - how many factor ANOVA and with what post-hoc test were used? 

RESPONSE: We were using one-factor ANOVA with only “health status” as an explanatory variable. However, as the reviewer kindly pointed out that this may be confusing, we changed to four-factor ANOVA so that the variables used are consistent over all of our analyses. This has now been clarified in the revised Material and Methods (page 4, lines 192-196). We also added table S5 to report our results. However, this did not change the overall outcome: health status did not explain alpha diversity (revised Results, page 6, lines 268-270).  

Line 225 - "background information of the IBD-horses" are not strictly results and should rather be moved to the description of the animals (line 122). 

RESPONSE: We agree with the reviewer and have now moved this chapter to the Material and Methods (page 3, lines 113-136). 

Reviewer 3 Report

Comments and Suggestions for Authors

IBD is an important issue in clinical science which is a growing problems. The gut microbiota of horse were checked and proposed to be used as marker for detecting IBD.

Following improvements are suggested

Line 15-16: You have not worked on humans and other animals. So, change this line with respect to horse.

Introduction

I found this section is very lengthy like a review paper. Short introduction with focused hypothesis is required to be written.

So, please reduce this section to

1.        About  IBD in animals, humans, and its international status.

2.        Gut microbiota as markers in other diseases and in IBD.

3.        Works on IBD such as doi:10.3389/fendo.2023.1217165 and the gap, followed by the hypothesis based on the gap.

Materials methods

Line 142: Magoč et al. 2011 need to be numbered

Section 2.4 may be presented before section 2.3

Results

Figure 2 is not legible, especially the text portions

Discussion

This part is ok

Conclusion

Please shorten the conclusion and include most of the important results

Author Response

REVIEWER 3 

IBD is an important issue in clinical science which is a growing problem. The gut microbiota of horse was checked and proposed to be used as marker for detecting IBD. 

Following improvements are suggested. 

Line 15-16: You have not worked on humans and other animals. So, change this line with respect to horse. 

RESPONSE: We have changed the sentence according to the reviewer’s suggestion (page 1, lines 16-18). 

Introduction 

I found this section is very lengthy like a review paper. Short introduction with focused hypothesis is required to be written. 

So, please reduce this section to 

1.        About IBD in animals, humans, and its international status. 

2.        Gut microbiota as markers in other diseases and in IBD. 

3.      Works on IBD such as doi:10.3389/fendo.2023.1217165 and the gap, followed by the hypothesis based on the gap. 

RESPONSE: We have now shortened the Introduction and moved some relevant parts in the Discussion. In addition to those three sections that the reviewer suggested we had to keep also the paragraph about fecal calprotectin concentration as a potential biomarker of horse IBD (as this was a part of the study) and shortly mention the use of fecal samples as material to study gut microbiota composition. We have also acknowledged the paper, the reviewer suggested, in the revised Discussion (page 10, lines 411-413). 

Materials and methods 

Line 142: Magoč et al. 2011 need to be numbered. 

RESPONSE: Thank you, that was a typing error. It has now been numbered (page 4, line 152). 

Section 2.4 may be presented before section 2.3 

RESPONSE: We agree with the reviewer, the order of these sections has now been switched.  

Results 

Figure 2 is not legible, especially the text portions. 

RESPONSE: We noticed that there was a small error in ANCOMBC analysis (we had not merged stallions (N=4) and geldings). We changed this in the revised Results (pages 6-8, several highlighted pieces of text) and re-did Figure 2. Due to re-analysis, we observed a reduced number of differentially abundant taxa and decided to include taxa altered by either survey- or acute-IBD status with FDR<0.1 to figure 2 (as now stated in the figure legend). Additionally, we increased the font size, changed the family colors so that families with single phylum now have similar colors, and improved the axis titles. The family colors are also updated in the complementary Supplementary figure.